


**Atmospheric mercury in the Southern Hemisphere – Part 2: Source**
**apportionment analysis at Cape Point station, South Africa.**
Johannes Bieser[1], Hélène Angot[2], Franz Slemr[31], Lynwill Martin[4]
Corresponding authors: johannes.bieser@hzg.de Lynwill.Martin@weathersa.co.za
[1]Helmholtz-Zentrum Geesthacht (HZG), Institute of Coastal Research, Max-Planck-Str. 1, D-
21502 Geesthacht, Germany
[2]Institute of Arctic and Alpine Research, University of Colorado Boulder, Boulder, CO, USA.
[3]Max-Planck-Institut für Chemie (MPI), Air Chemistry Division, Hahn-Meitner-Weg 1, D-55128
Mainz, Germany
[4]South African Weather Service c/o CSIR, P.O.Box 320, Stellenbosch 7599, South Africa

# 13 Abstract

Mercury (Hg) contamination is ubiquitous. In order to assess its emissions, transport,
atmospheric reactivity, and deposition pathways, Hg monitoring stations have been
implemented on a global scale over the past 10-20 years. Despite this significant step forward,
the monitoring efforts have been insufficient to fulfill our understanding of Hg cycling in the
Southern Hemisphere. While oceans make up 80% of the Southern Hemisphere's surface
area, little is known about the effects of oceans on Hg cycling in this region. For instance, in
the context of growing interest in effectiveness evaluation of Hg mitigation policies, the relative
contribution of anthropogenic and legacy emissions to present-day atmospheric Hg levels is
unclear. This paper constitutes Part 2 of the study describing a decade of atmospheric Hg
concentrations at Cape Point, South Africa, i.e. the first long-term (> 10 years) observations in
the Southern Hemisphere. Building on the trend analysis reported in Part 1, here we combine
atmospheric Hg data with a trajectory model to investigate sources and sinks of Hg at Cape
Point. We find that the continent is the major sink and the Ocean, especially warm regions, is
the major source for Hg.
Further, we find that mercury concentrations and trends from long range transport are
independent of the source region (e.g. South America, Antarctica) and thus indistinguishable.
Therefor, by filtering out air masses from source and sink regions we are able to create a
dataset representing a southern hemispheric background Hg concentrations. Based on this
dataset we were able to show that the inter-annual variability of Hg concentrations is not driven
by changes in atmospheric circulation but rather due to changes in global emissions (gold
mining and biomass burning).



# 35 1. Introduction

Mercury (Hg) is a toxic pollutant that is ubiquitous in the environment. Due to anthropogenic
emissions, the amount of mercury in the atmosphere has increased sevenfold since pre-
industrial times (Amos et al., 2013). Mercury occurs in the atmosphere as gaseous oxidized
mercury (GOM), particle bond mercury (PBM) and predominantly as gaseous elemental
mercury (GEM). Because of its atmospheric lifetime of about 1 year, once emitted into the
atmosphere, GEM is transported on hemispheric and global scales (Slemr et al., 2018). Since
2017 usage and emissions of Hg are regulated under the UN Minamata Convention on
Mercury (UNEP, 2013). This UN convention forces its member states to assess the current
state of mercury pollution, take actions to reduce mercury emissions, and to evaluate the
success of these measures on a regular basis.
In order to assess the impact of emission reductions on the system it is necessary to better
understand the sources and sinks driving atmospheric mercury cycling. Especially in the
southern hemisphere there has been a lack of long-term atmospheric observations that allow
to investigate and distinguish long-term trends from the natural variability of atmospheric Hg
concentrations. So far, the only long-term observations in the southern hemisphere with
measurements over more than 10 years have been and are performed at Cape Point (CPT),
South Africa, where Hg has been measured since 1995 (Baker et al., 2002, Slemr et al., 2008).
At CPT, for the first ten years (September 1995 to December 2004) Hg concentrations showed
a decreasing trend (Slemr et al., 2008, Martin et al., 2017) while Martin et al. (2017) identified
an increasing trend for the last ten years (March 2007 to June 2015). Yet the reason for the
observed trends is unclear and there was no explanation for the change in sign from a
decreasing to an increasing trend.
This work is presented in two accompanying papers where the first one (Slemr et al.,
submitted) focuses on long-term trends in the southern hemisphere over the last ten years
based on measurements at CPT and Amsterdam Island (AMS) which is operational since
2012. The key finding of that paper is that since 2007 mercury concentrations at CPT seem to
have been increasing while no significant trend was found in the 2012 – 2017 period both at
CPT and AMS. The upward CPT trend in 2007 – 2017 period seems to be driven by
exceptionally low Hg concentrations in 2009 and above average concentrations in 2014.
Here, we combine ten years of Hg (2007 – 2016) observations at CPT with calculated hourly
backward trajectories in order to investigate sources and sinks for mercury and to quantify the
impact of long-term changes in atmospheric circulation patterns on observed Hg
concentrations at CPT. The aim of this study is to:
- distinguish between local changes at CPT and hemispheric Hg trends;
- identify source and sink regions for Hg at CPT;
- estimate the natural variability of Hg concentrations at CPT in order to distinguish them
from other effects such as changing emissions.



This paper aims to improve our understanding of mercury cycling in the southern hemisphere.
For this, we elaborate on the research question whether concentrations and trends observed
at CPT are dominated by local signals or representative for mercury cycling across large parts
of the southern hemisphere. Based on backward trajectories and statistical modeling we
investigate source and sink regions for mercury observed at Cape Point and the impact of
inter-annual variability on atmospheric transport patterns and emissions processes.

# 79 2. Methodology

## 80 2.1 Observations

This study is based on ten years (2007-2016) of continuous gaseous elemental Hg
measurements at Cape Point (CPT, 34º21′S, 18º29′E), South Africa. The CPT measurement
site is part of the GAW (Global Atmospheric Watch) baseline monitoring observatories of the
World Meteorological Organization (WMO). It is located at the southernmost tip of the Cape
Peninsula on top of the cliff at an altitude of 230 m (a.s.l.). There are no major local Hg sources
and the nearest city, Cape Town, is located 60 km to the north (see Fig. 1). The station is in
operation since the 1970ties and, besides Hg, several other pollutants are measured on a
regular basis. These include $CO_2$, CO, ozone, methane, and radon ($^{222}$Rn) which we use to
substantiate the findings on mercury. A detailed description of the CPT station can be found in
the accompanying paper (Slemr et al., submitted).

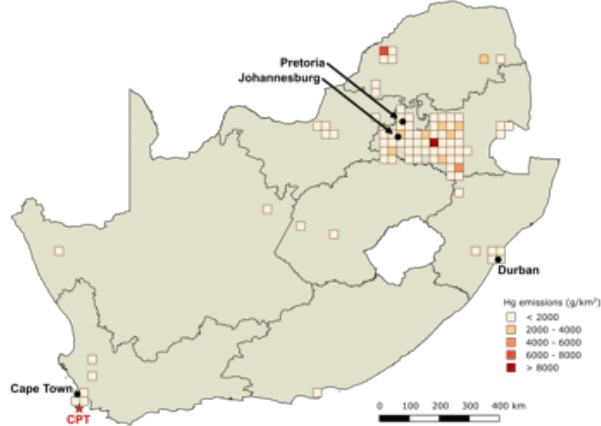

*Figure 1: Location of the Cape Point site (CPT, red star), at the southernmost tip of the Cape Peninsula,*
*and of known anthropogenic mercury emission sources (in g/km², Global Mercury Assessment 2018*
*emission inventory) in South Africa. This map was made with QGIS.*





## 2.2 Modeling

GEM measurements (gaseous elemental Hg, the dominant form of Hg in the atmosphere (~95%)) at CPT are performed continuously with a 15 min sampling interval. The GEM measurements were aggregated to hourly averages and for each hourly measurement an ensemble of 5-day backward trajectories was calculated using the HYSPLIT model (Stein et al., 2015) (Fig. 2). For the hourly trajectory ensembles we used different starting altitudes in order to capture the model uncertainty due to the model's initial conditions. The HYSPLIT model was run for ten years (2007 to 2016) using GDAS (Global Data Assimilation System) 0.5°x0.5° degree meteorological inputs based on the NCEP/NCAR reanalysis dataset (Kalnay et al., 1996, NOAA, 2004).

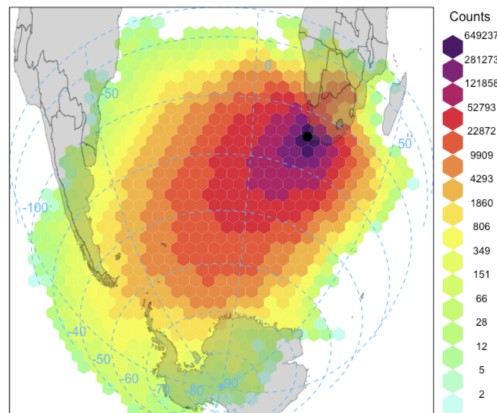

*Fig. 2: Origin of air masses influencing the Cape Point site (black dot). Gridded back trajectory frequencies using an orthogonal map projection with hexagonal binning. The tiles represent the number of incidences. 2007-2016 hourly back trajectories were computed using the HYSPLIT model (Stein et al. 2015) and the figure was made using the R package openair (Carslaw and Ropkins, 2012).*

## 2.3 Regionalization

The trajectories were categorized into six source regions depending on their travel path (Table 1). These categories are:

- **Local**

    Air parcels which traveled less than 100 km absolute distance to CPT over the last three days are considered to be local air masses.

- **Continental**

    Air parcels that spend more than 80% of travel time over the African continent.

- **Eastern Ocean**



Air parcels which did not travel over land and did not go west of 30° E within the
last 4 days.

-   **South American**

Air parcels which were west of 30°W within the last 4 days.

-   **Antarctic**

Air parcels which were south of 55° S within the last 4 days.

-   **Atlantic**

Air masses which do not fall within the other categories and spend more than
80% of the time over the Atlantic Ocean. This category makes up the majority of
all trajectories.

Naturally, the categorization of air parcels depends on the definition of regions of origin and the
travel time chosen for the algorithm. We calculated 5-day backward trajectories and
experimented with different cutoff values to determine the source regions of the air masses
(Table 1). For this study we chose a cutoff time of 4 days to determine long range transport
from Antarctica and South America. However, he choise of cuttoff times of 3 or 5 days did not
change the conclusions of our study. This decision is based on tests with different cutoff times
and on the fact that the uncertainty of the trajectories grows with travel time. Moreover, air
parcels are often a mixture of different source regions (e.g. Atlantic/continental). As an
additional test for the calculated categorization we used secondary parameters such as $^{222}$Rn,
CO, $CO_2$, $CH_4$, and $O_3$. $^{222}$Rn is a radioactive gas of predominantly terrestrial origin with a half-
life of 3.8 days. Thus, high $^{222}$Rn concentrations mark air masses which recently passed over
the continent such as "Continental" and "Local". Other examples are the distinction of long-
range transport from South America from Atlantic air masses. Here, we would also expect
higher concentrations of other anthropogenic pollutants (e.g. $^{222}$Rn, $CH_4$).

| Days | Antarctic | S. America | Continental | Eastern O. | Atlantic | Local |
|---|---|---|---|---|---|---|
| 2 | 1778 (1%) | 150 (< 1%) | 14580 (9%) | 5930 (4%) | 143478 (86%) | 1760 (1%) |
| 3 | 11800 (7%) | 3614 (2%) | 10596 (6%) | 7842 (5%) | 132876 (79%) | 926 (1%) |
| 4 | 26696 (16%) | 12756 (8%) | 7928 (5%) | 7882 (5%) | 111770 (67%) | 550 (<1%) |
| 5 | 39710 (24%) | 22960 (14%) | 5906 (4%) | 6876 (4%) | 91666 (55%) | 370 (<1%) |

*Table 1: Impact of travel time cutoff on air parcels source region categorization.*



## 144 2.4 Identification of source/sink regions

In order to evaluate source and sink regions we calculated the 10[th] and 90[th] percentile of GEM
measurements for each season (seasons being defined as three-month intervals: DJF
(summer), MAM (autumn), JJA (winter), SON (spring). This seasonal filter proved to be
necessary to remove the annual cycle in GEM concentrations driven (among others) by the
seasonality of emissions, planetary boundary layer height, and transport patterns.
Furthermore, we filtered out mercury depletion events, of unknown origin (Brunke et al. 2010),
which were defined as hourly average GEM concentrations of less 0.25 ng/m³.
For the source/sink region analysis we interpolated hourly trajectory locations onto a polar
stereographic grid centered over the South Pole and calculated the total amount of trajectories
traveling through each grid cell over the ten year (2007-2016) time span. We then performed
the same procedure for the trajectories of the 10[th] and 90[th] percentile GEM concentrations. By
dividing these percentile maps by the total amount of trajectories traveling through each grid
cell, we created maps indicating the regional prevalence of high and low GEM concentrations.
In the theoretical case of perfectly homogeneous, evenly distributed sources and sinks each
grid cell would have a value of 0.1 indicating that 10% of all air parcels in each grid cell belong
to the 10% highest/lowest GEM observations. Deviations from this uniform distribution are then
interpreted as source regions for high/low GEM concentrations. E.g. a value of 0.2 indicates
that twice as many high/low GEM concentrations originate from a given grid cell compared to a
uniform distribution.
To better distinguish the 10th and 90th percentile plots we chose opposite color schemes for
the 10th and 90th percentile plots. In the case of the 90th percentile plots red color indicates
source regions for high GEM concentrations (i.e. > 0.1) while blue color indicates the absence
of sources in this region (Fig 3a). For the 10th percentile plots blue color indicates sink regions
for GEM concentrations (Fig 3b) while red color indicates the absence of sinks. It is important
to note that an absence of sources is not equal to the presence of sinks and vice versa. Figure
3 gives an example of these plots for air masses attributed to the 'Atlantic' category for ²²²Rn
measurements. This plot serves as an evaluation of the regionalization algorithm. It can be
seen that high ²²²Rn concentrations are found only in air masses that traveled over the
continent (Fig 3a). Similarly, Figure 3b depicts the fact that no measurements with low ²²²Rn
concentrations were found in air masses that traveled along the coast line, indicating an impact
of anthropogenic sources. Finally, this procedure is sensitive to the total amount of trajectories
traveling through a grid cell which leads to low signal to noise ratios in the outskirts of the plot
where only few trajectories originate at all. We used a cutoff value of 10 hits and discarded all
grid cells with fewer hits but this still leads to a few non-significant hot spots at the outskirts of
the domain (e.g. Fig 3a in Antarctica).





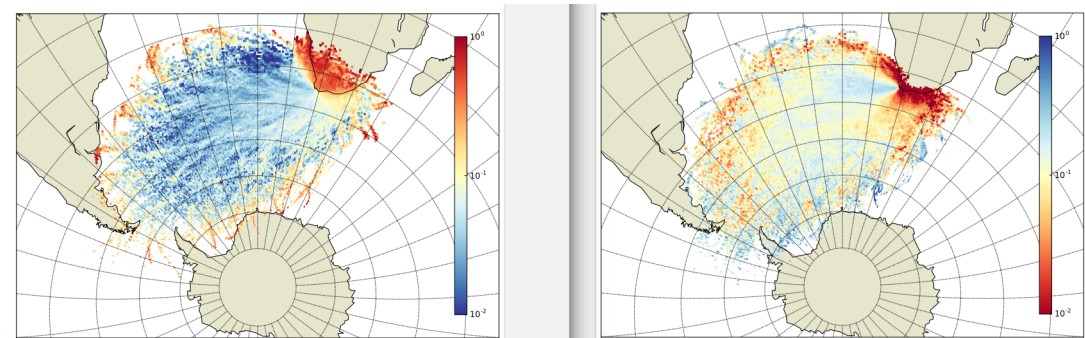


*Figure 3: Distribution map for the 90th percentile highest [222]Rn concentrations (left) and the 10th percentile lowest [222]Rn concentrations (right) measured at Cape Point. Values are the dimensionless prevalence of air parcels of a given concentration percentile ranging from 0 to 1.*

# 3. Results

## 3.1 Source and sink regions

To gain an initial overview we investigate 10th and 90th percentile maps for all GEM measurements over the whole period 2007-2016 (Fig. 4). It can be seen that low GEM concentrations originate almost exclusively from air masses which traveled over the continent (Fig 4b). This result is in line with a cluster analysis performed by Venter et al. (2015) *"Air masses that had passed over the very sparsely populated semi-arid Karoo region, almost directly to the north of CPT GAW, were mostly associated with [...] lower GEM values"*. It is also consistent with the finding of Slemr et al. (2013) that southern Africa, based on Hg vs [222]Rn correlations, is a net sink region. The reason for this is probably a mixture of near zero emissions in the region and increased dry deposition on the surface.

Over the Atlantic Ocean, low GEM concentrations are in line with a uniform distribution with values mostly only slightly below the equilibrium value of 0.1. The exception are air masses that travelled over the ocean east of Cape Point where almost no low concentration GEM measurements originated. Looking at the highest 90th percentile of GEM concentrations, air masses travelling over the ocean show a lower abundance with the exception of a patch east of Cape Point (Fig 4a).

The picture becomes clearer when plotting trajectories independently for each of the previously defined regions (Fig. 5). It can be seen that air masses from the eastern ocean sector are the predominant source region of air masses with elevated GEM concentrations (Fig. 5e). In this region the Agulhas Current transports warm water from the Indian Ocean to the Atlantic Ocean. For continental air masses (Fig. 5b) certain source regions can be identified. These coincide with known major Hg emitters (Fig. 1). For air masses representing long range transport (Atlantic, South American, Antarctic) frequency values of the 90th percentile highest GEM concentrations are mostly around 10% indicating no specific sources or sinks in these regions.





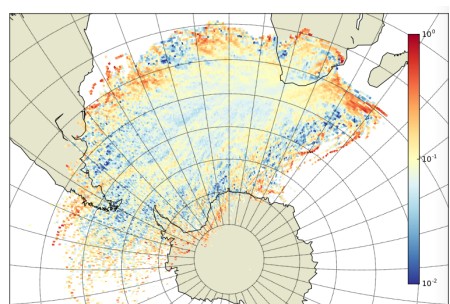
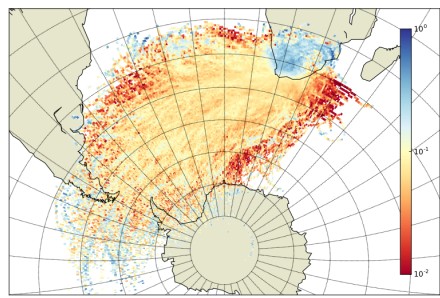

Figure 4: Prevalence of highest (90th percentile) (left) and lowest (10th percentile) (right) GEM concentrations using all hourly trajectories over ten years.

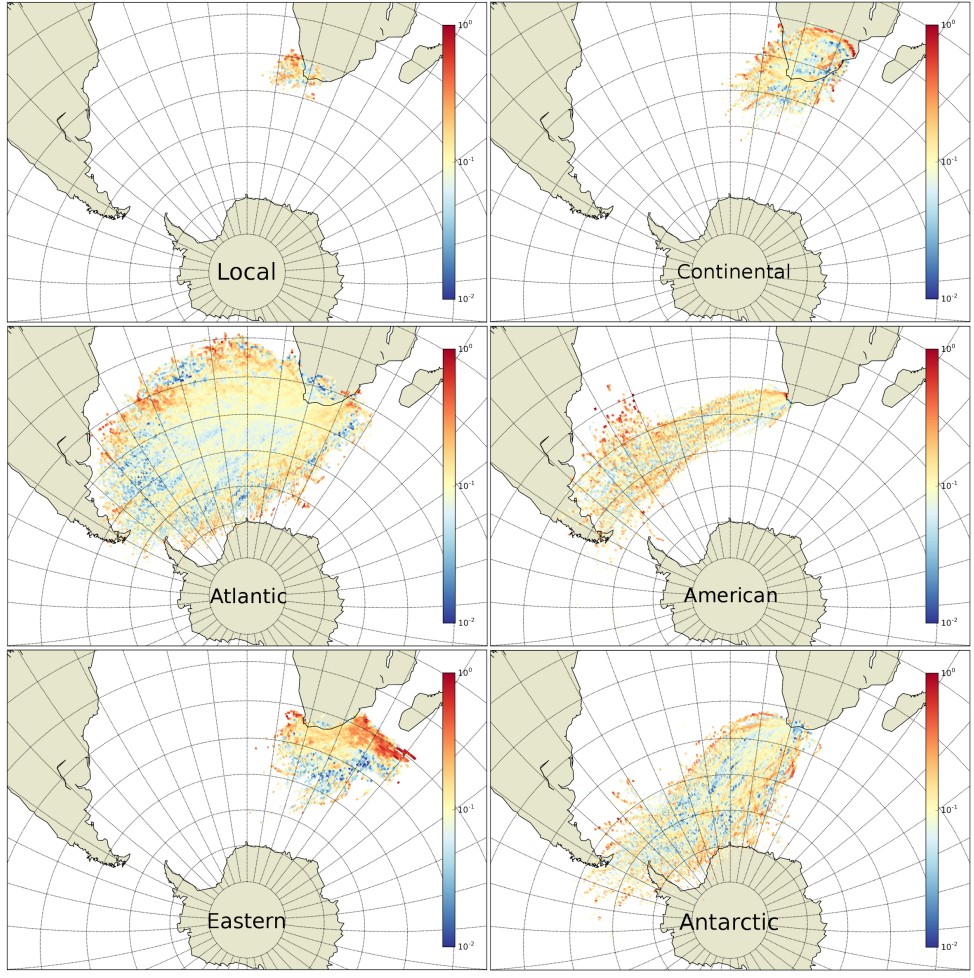

Figure 5: 90th percentile highest GEM concentrations for air masses from six source regions. Red color means source regions, blue one absence of emissions in that region.





Looking at the 10th percentile of lowest GEM concentrations, regional and continental air
masses can be identified as the single most important sink region (Fig. 6a,b). There are also
some continental areas with a high prevalence of low Hg concentrations attributed to the
Atlantic sector. These can be interpreted as air parcels with a mixed continental/Atlantic travel
path that have been attributed as Atlantic air masses by the algorithm as they did not spend
enough time over the continent to be attributed to this sector. Finally, there are no air masses
with low GEM concentrations originating from the eastern ocean sector (Fig. 6e). Yet again,
there is no clear picture concerning air masses from long range transport.

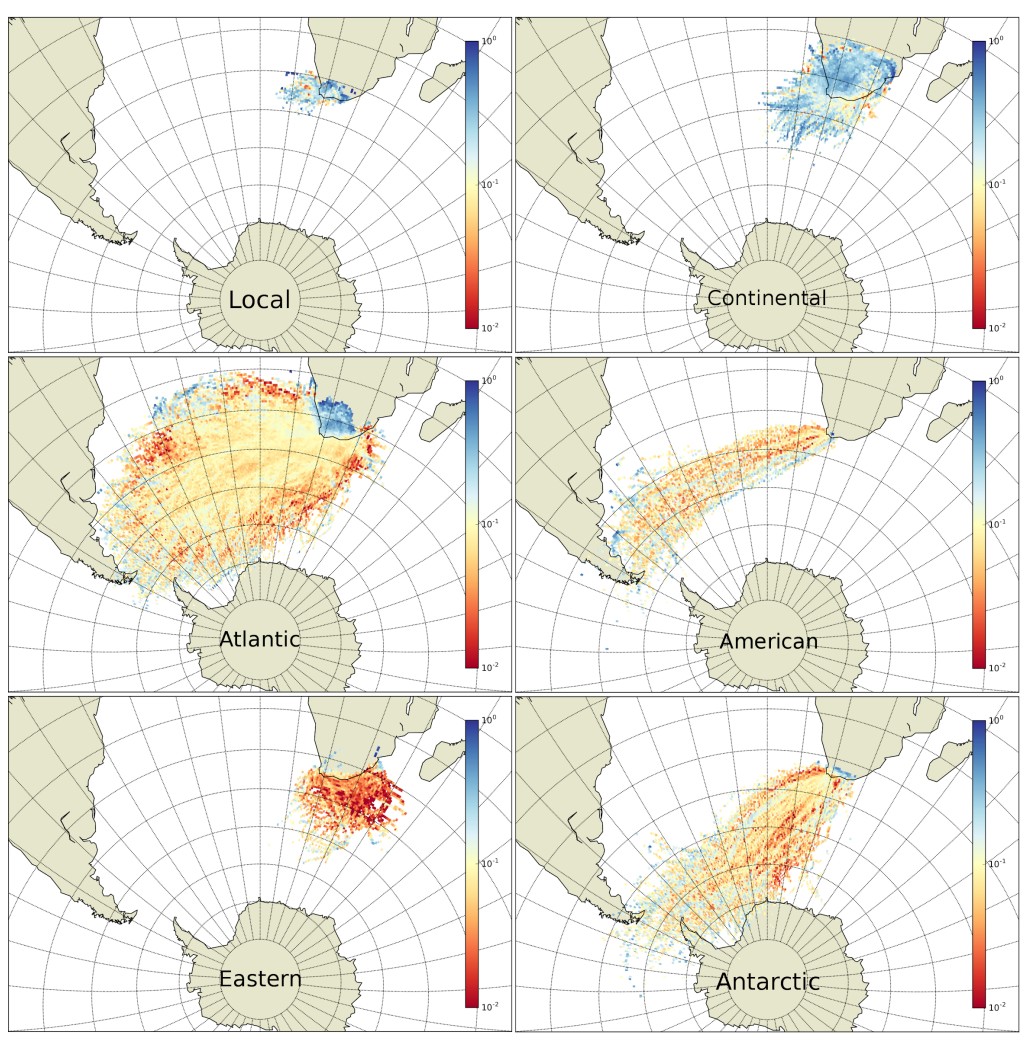

*Fig. 6: 10th percentile highest Hg concentrations for six source regions. Blue colors indicate sink*
*regions, red color absence of sinks. Low values are almost exclusively linked to continental and local air*
*masses.*





To further investigate the processes behind the observed source and sink regions, we regard
seasonal trajectory maps (Fig. 7). This analysis reveals, that the high GEM concentrations
associated to air masses from the eastern ocean occur mainly during austral spring and
summer. This indicates that temperature or primary production and a related increase in
evasion of GEM (i.e., reemission of legacy Hg) from the ocean might explain these
observations.

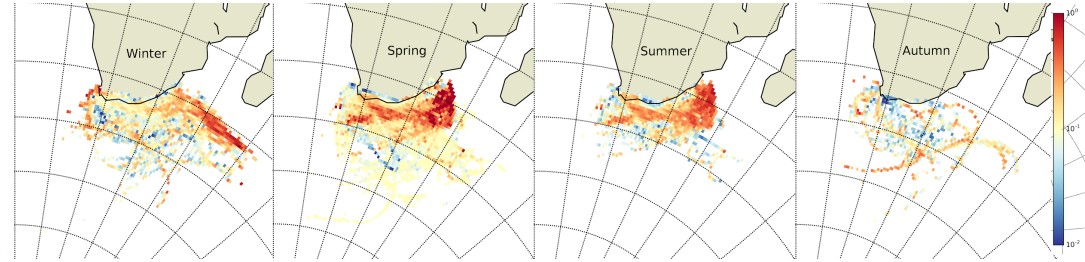

*Figure 7: Seasonal breakdown of 90th percentile highest Hg concentrations from the eastern ocean*
*source region. Highest Hg concentrations are mostly associated to Austral spring and summer which*
*supports primary production as a potential source for oceanic Hg releases.*

## 237 3.2 Comparison of regionalized data

Annual and monthly averages and medians for each source region of 4-days backward
trajectory were calculated for GEM, $CO_2$, $^{222}Rn$, CO, $CH_4$, and $O_3$. Here we compare the annual
averages and medians for all species and discuss the implications this comparison provides on
regionalization.
The highest annual average and median GEM concentrations were found for "Eastern Ocean"
in all years except for 2015. The lowest annual GEM concentrations were almost always either
"Local" (averages in 2007, 2010, 2013; medians in 2007, 2010, 2013, 2014) or "Continental"
(averages in 2008, 2009, 2012, 2016; medians in 2008, 2010, 2013, 2015). Annual average
and median GEM concentrations for "South American", "Antarctic" and "Atlantic" lie in the
middle in varying order.
The highest and second highest $^{222}Rn$ annual average and median concentrations were found
for "Continental" and "Local", respectively, in all years except in 2007, in which the "Local"
concentration is larger than the "Continental" one. This is consistent with the terrestrial origin of
$^{222}Rn$, a radioactive gas with a half-life of 3.8 days (Zahorowski et al., 2004). The lowest and
second lowest $^{222}Rn$ concentrations were found in air masses attributed to "South American"
and "Antarctic" (averages in all years except 2009, 2010, and 2011; medians in all years
except 2010 and 2011), respectively. In the exceptional years the lowest $^{222}Rn$ concentrations
were "Antarctic" and the second lowest "South American". The concentrations attributed to
"Eastern Ocean" and "Atlantic" lie in the middle, with "Eastern Ocean" concentrations being
mostly higher than the "Atlantic" ones (always in medians, in averages all years with exception
of 2009 and 2010).
The highest and second highest annual CO mixing ratios were found in air masses attributed
to "Continental" and "Local", respectively, with a few years with reverse order (averages: 2007-
2009; medians: 2007 and 2016). The lowest annual CO mixing ratios were found in air masses





attributed to "Antarctic" (averages in 2007, 2009, 2011, 2013, 2015; medians in 2007, 2009,
2011, and 2014) or "South American" (averages in 2008, 2012, 2014, 2016; medians in 2012
and 2016). CO mixing ratios for "Atlantic" and "Eastern Ocean" lie in the middle with varying
order.
The highest annual average and median $CH_4$ mixing ratios were observed in air masses
attributed almost always to "Local" or "Continental" (averages "Local" in 2007, 2008, 2011,
2014, 2016 and "Continental" in 2009, 2010, 2012, 2013, 2016; medians "Local" in 2007, 2010,
and 2014 – 2016 and "Continental" in 2008, 2009, and 2011 - 2013). Mixing ratios "Eastern
Ocean" were mostly the lowest (averages in 2007, 2009-2012, 2015; medians in 2007, 2009 –
2011, 2015) or second lowest (averages in 2008 and 2013; medians in 2008, 2012, 2016).
Annual mixing ratios for "South American", "Antarctic" and "Atlantic" lie mostly in the middle
with varying order.
The highest average and median annual CO2 mixing ratios are mostly attributed to air masses
of either "Local" or "Continental" origin with only a few exceptions (average in 2007; median in
2012). The lowest annual mixing ratios are mostly either "Antarctic" or "South American" with
only a few exceptions (average and median in 2012). Annual averages and medians for
"Atlantic" and "Eastern Ocean" are mostly in the middle.
$O_3$ is being photochemically produced in NOx rich air masses, i.e. mostly in air masses of
continental origin, and destroyed in NOx poor air masses, i.e. mostly over the remote ocean
(Monks et al., 1998; Fischer et al., 2015). As expected, annual "Continental" average and
median $O_3$ mixing ratios are highest or second highest (averages highest in 2007, 2009 –
2011, 2013 – 2015, second highest in 2008, 2012, and 2016; medians highest in 2007, 2009,
2010, and 2014, second highest in 2008, 2011, 2012, 2015 and 2016). The lowest annual
average and median $O_3$ mixing ratios are attributed to "Eastern Ocean" in all years except
2008 and 2014, in which the mixing ratios are second lowest. The second lowest annual
average and median $O_3$ mixing ratios are found in air masses attributed to "Atlantic" (averages
and medians in 2007, 2009, 2011, 2013, 2015). An exception are the years 2008 and 2014 in
which "Atlantic" provides the lowest mixing ratios.
The results reported above apply for regionalization using 4-days backward trajectories.
Regionalization with 3 or 5-days backward trajectories provides similar results.
In summary, regionalized average and median annual mixing ratios of CO, $CO_2$, $CH_4$, and
especially concentrations of $^{222}$Rn behave as expected for species of terrestrial origin, i.e. with
highest values almost always in air masses attributed to "Continental" or Local" and the lowest
ones attributed mostly to "Antarctic" or "South American". "Atlantic" and "Eastern Ocean" tend
to lie in the middle. Ozone, although not of terrestrial but of photochemical origin in NOx rich
environments, also fits this pattern because its mixing ratios are highest in "Local" or
"Continental" air masses where the highest NOx mixing ratios are expected. The lowest $O_3$
mixing ratios in "Eastern Ocean" and "Atlantic" can be perhaps explained by NOx poor
environment as in "Antarctic" and "South American" but larger photochemical destruction due
to lower latitude and thus higher solar radiation. GEM, with highest concentrations in air
masses attributed to "Eastern Ocean" and the lowest one found either in "Local" or
"Continental", behaves just in contrary fashion to all the species mentioned above. Its contrary
behaviour pattern clearly shows that its sources are predominantly oceanic and the sinks
terrestrial.





## 3.3 Regional trends

Regionalised trends for GEM, $CO_2$, $^{222}Rn$, CO, $CH_4$, and $O_3$ were calculated from regional monthly averages and medians using least square fit. Months with less than 10 measurements were not considered. The trends of $^{222}Rn$ and $O_3$ are insignificant for all regions. For the remaining species we present only trends significant both in monthly averages and medians. The trend differences are tested for significance by comparison of averages (Kaiser and Gottschalk, 1972) using the slope and its uncertainty as an average and its standard deviation, respectively.

$CO_2$ and $CH_4$ trends are significant for all regions. The regional $CO_2$ trends are all significantly different (>99.9%) from each other with the exception of the trends of monthly medians of "Eastern Ocean" and "Atlantic". The highest trend was observed in "Continental" air masses (averages 2.24 ± 0.04 ppm $yr^{-1}$, medians 2.23 ± 0.04 ppm $yr^{-1}$) and the lowest in "Local" (averages 2.07 ± 0.12 ppm $yr^{-1}$, medians 2.10 ± 0.11 ppm $yr^{-1}$), the latter with exceptionally high uncertainty. The sequence of the remaining regions is "Eastern Ocean" > "Antarctic" > "Atlantic" > "South American" for trends from monthly averages and "Antarctic" > "Atlantic" > "Eastern Ocean" > "South American" for trends from monthly medians.

The highest $CH_4$ trends were observed in "Continental" in monthly averages (7.52 ± 0.76 ppb $yr^{-1}$) with "Eastern Ocean" being the second highest, and "Eastern Ocean" in monthly medians (7.34 ± 0.55 ppb $yr^{-1}$) with "Continental" being the second highest. The lowest trends were found to be in "South American" both in monthly averages and medians (averages 6.21 ± 0.54 ppb $yr^{-1}$, medians 6.37 ± 0.47 ppb $yr^{-1}$). The trends between these extremes were mostly not significantly different. The only significant CO trend both in monthly averages and medians was found for "South American" air masses (averages -1.30 ±0.58 ppb $yr^{-1}$, medians -1.02 ± 0.42 ppb $yr^{-1}$) and, therefore, cannot be compared with trends for other regions.

Three source regions provide significant trends for GEM, both in averages and medians. The trends for "Antarctic" and "South American" air masses are comparable (averages "Antarctic" with 14.25 ± 2.97 pg $m^{-3}$ $yr^{-1}$, "South American" with 14.88 ± 3.95 pg $m^{-3}$ $yr^{-1}$; medians "Antarctic" with 12.72 ± 2.86 pg $m^{-3}$ $yr^{-1}$, "South American" with 13.46 ± 3.96 pg $m^{-3}$ $yr^{-1}$). They are both almost twice as large as the trend for "Atlantic" air masses (averages 8.69 ± 2.54 pg $m^{-3}$ $yr^{-1}$, medians 8.49 ± 2.52 pg $m^{-3}$ $yr^{-1}$). This indicates that they are representative of the SH background.

Regionalisation with 4-days backward trajectories provide only a few monthly values for "Local" because less than 1% of measurements could be attributed to this class. The trends for the remaining regions provide a similar pattern as regionalisation with 3 or 5 days backward trajectories. The highest and lowest $CO_2$ trends were found for "Continental" and "South American", respectively, both when calculated from monthly averages and medians. The highest $CH_4$ trends were observed for "Continental" when calculated from monthly averages and for "Eastern Ocean" when calculated from monthly medians. The second highest $CH_4$ trends were for "Eastern Ocean" in the former and "Continental" in the latter case. The lowest and second lowest $CH_4$ trends were found for "Antarctic" and "South American" air masses. Similarly to 3-days regionalisation, the GEM trends for "Antarctic" and "South American" were comparable and significantly larger than those for "Atlantic". Only the difference between the former two and the latter one is smaller than in the 3-days regionalisation.





In summary, the patterns of GEM, $CO_2$, and $CH_4$ trend differences provide another piece of
evidence for an oceanic GEM source.

## 3.4 Regional abundance

In order to determine the impact of regional source and sink regions on mercury
concentrations at Cape Point and to evaluate whether observations at this location are
representative for the southern hemisphere in general, we investigate the abundance of air
masses from different source and sink regions and their impact on observed average
concentrations and long-term trends. Air masses from long range transport (Atlantic, Antarctic,
South American) make up 90% of all air masses observed at Cape Point. Seasonally averaged
observed concentrations from these regions show a high correlation with the averages of all
observations at Cape Point with $R^2$ values mostly above 0.9 (Table 2). Only Antarctic air
masses during austral summer and autumn exhibit a lower correlation.
Air masses from the sectors eastern ocean and continental on the other hand show very low
correlations with the averages observed at Cape Point indicating that these air masses differ
significantly from the rest. On average, transport from these two regions make up 10% of the
air masses at Cape Point (Table 1). Their prevalence varies mostly only by 1 to 2 percentage
points from year to year with a peak of 10% continental air masses in 2011. However, we
found that the prevalence of air masses from source and sink regions is not the driver of the
inter-annual variability of Hg concentrations at Cape Point. (e.g. even with twice as much as
average air masses from the sink region 2011 was no year with particularly low Hg
concentrations) (Figs. 7, 8). Because of this and based on the comparison with measurements
at Amsterdam Island (see accompanying paper) we are confident that mercury concentrations
observed at Cape Point are representative for the southern hemisphere background.
Additionally, based on the presented work we are able to filter out the source and sink regions
from the dataset for further analysis. Figure 9 depicts the whole GEM dataset with values from
source and sink regions highlighted.

|  | Annual | Spring (SON) | Summer (DJF) | Autumn (MAM) | Winter (JJA) |
|---|---|---|---|---|---|
| Antarctic | 0.89 | 0.95 | **0.75** | **0.72** | 0.96 |
| South American | 0.95 | 0.95 | 0.91 | 0.95 | 0.97 |
| Continental | **0.39** | **0.54** | **0.05** | **0.59** | **0.33** |
| Eastern Ocean | 0.81 | 0.77 | **0.58** | **0.14** | 0.84 |
| Atlantic | 0.98 | 0.97 | 0.90 | 0.94 | 0.99 |

*Table 2: Correlation coefficient ($R^2$) of regional average concentrations with averages of all*
*measurements at Cape Point. Values are based on monthly averages (N=30). Antarctic, Atlantic, and*
*South American air masses exhibit a high correlation with the overall mean concentrations observed at*
*Cape Point.*

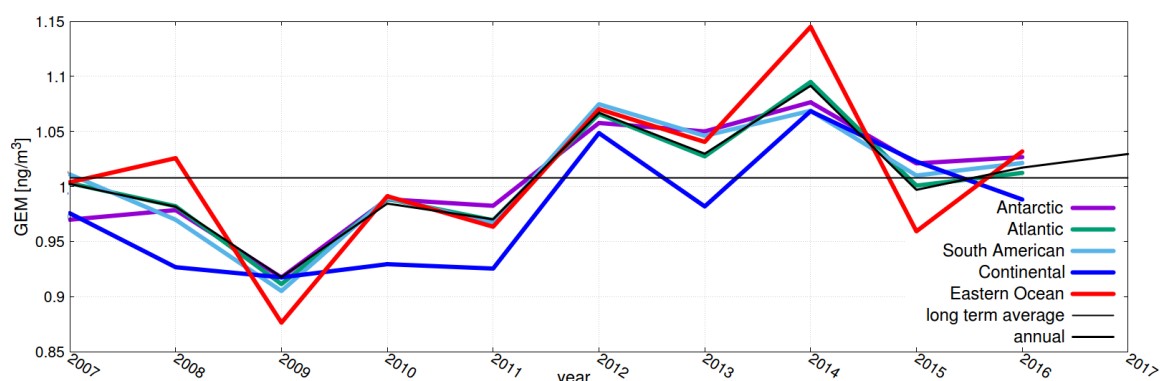

*Figure 8: Annual average concentrations at Cape Point from 2007 to 2017 (black line) and regional averages (colored lines). It can be seen that the minimum in 2009 and the maximum in 2014 is present in all source regions.*

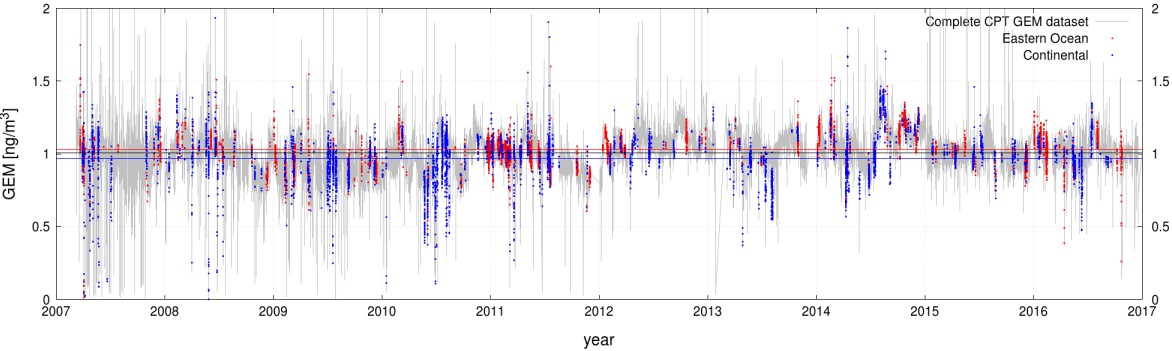

*Fig. 9 Complete Cape Point dataset (grey) with observations originating from source (red) and sink (blue) regions superimposed. The colored x-axis parallel indicates the long-term average (black: complete dataset, red: source region, blue: sink region).*

## 3.5 Inter-annual variability

So far the trend in GEM concentrations observed at Cape Point and the fact that it seemingly changed from increasing to stable between 2006 and 2017 is still unexplained. Having shown that the observations at Cape Point are not dominated by regional processes the question arises which large-scale processes modulate the signal on annual and decadal time scales.

At this point, our null hypothesis is that mercury concentrations in the southern hemisphere were stable over the last decade but processes on global and hemispheric scales superimpose a (multi-)annual modulation on the signal. Based on our analysis so far we can exclude changes in atmospheric transport patterns as the cause for the inter-annual variability and correct the data for local source and sink regions. Thus, in our opinion only global source processes remain as possible explanations for the observed anomaly. The identified processes





are: marine emissions, biomass burning and gold mining which are the major sources for
mercury in the Southern Hemisphere, as well as volcanic activity.
Especially, the low mercury concentrations observed in 2009 and the high values observed in
2014 seem to be at least partially a large-scale phenomenon. A screening of international
observation networks showed that also Mace Head – which is located in the Atlantic Ocean in
the northern hemisphere – also has the lowest annual average mercury concentrations in 2009
and the highest in 2014. For the year 2009 the mercury emission inventory of Streets et al.
(2019) postulates a sudden plummet in global gold mining activity. Comparing the annual
anomaly from the ten-year average, gold mining activity are correlated with observed GEM
concentrations (R=0.64). Similarly, we found a correlation with biomass burning in the
Southern Hemisphere (mostly Africa) (R=0.75) (Jiang et al., 2017). We removed air masses
from the identified source and sink regions from the data set and used a regression analysis to
correct for changes in global gold mining and biomass burning emissions (Figure 10). The
resulting signal becomes relatively flat with only two peaks remaining in 2012 and 2014. We
hypothesize that volcanic emissions which were above average in 2011 and 2014 might be
responsible for higher mercury concentrations in those years.

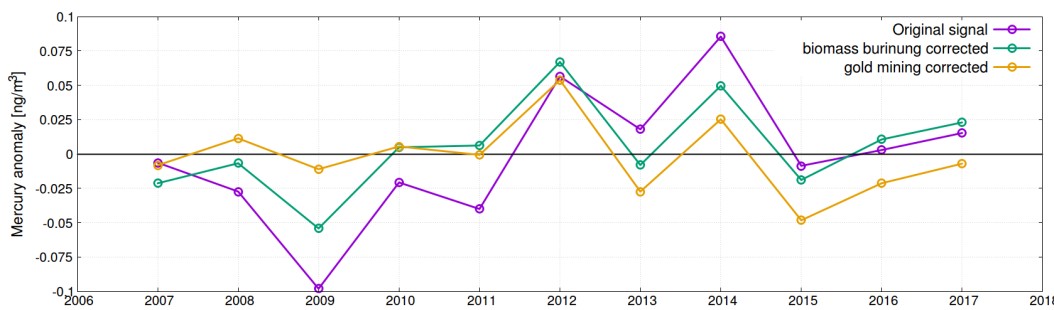

*Fig. 10: Annual anomaly from ten-year average mercury concentrations. Original dataset at Cape Point*
*(purple), corrected for biomass burning emissions (green) (Jiang et al., 2017) and additionally corrected*
*for gold mining emissions (orange) (Streets et al., 2019).*

# 417 4. Conclusions

Our goal was to improve the understanding of mercury cycling in the Southern Hemisphere.
For this, we combined ten years of GEM observations at Cape Point, South Africa with hourly
backward trajectories calculated with the HYSPLIT model. Our findings are that:
(1) The continent is the major sink region for mercury with the exception of significant point
sources, mostly linked to coal combustion.
(2) The warm Agulhas current to the south-east is the major source of atmospheric mercury
observed at Cape Point.





(3) Separating the ground-based observations into air parcels from different source regions
showed that mercury behaves opposite to known pollutants of anthropogenic and continental
origin.
(4) Mercury concentration in air masses from Antarctic, Atlantic, and south American origin
were statistically indistinguishable. We interpret these observations as a good representation
of the southern hemispheric background.
(5) We find that the trends in GEM concentrations postulated in the past are probably an
artifact of single years with unusually high or low GEM concentrations (see accompanying
paper, this SI). We were able to show that these exceptional years could be explained by
changes in global emissions from biomass burning and gold mining, two major sources of
mercury in the Southern Hemisphere.
(6) With the Ocean as the main source of mercury in the southern hemisphere it can be
expected that an increased air-sea flux due to the larger concentration gradients will
compensate reductions in global atmospheric emissions in the short term future.

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
