# Peer review of "Atmospheric mercury in the Southern Hemisphere – Part 2: Source apportionment analysis at Cape Point station, South Africa."

_Atmospheric Chemistry and Physics, 2020_

## Referee Comment (RC1) · Anonymous Referee #1 · 19 Mar 2020

General comments

The manuscript reveals the source and sink of atmospheric gaseous elemental mercury (GEM) arriving at the Cape Point based on the in-depth back trajectory analysis. The results obtained by the authors have the interest and contribute for understanding of Hg behaviors in the global atmosphere and present a defensible dataset and interpretation, building on previous studies. The manuscript is well-organized, and the methods utilized are appropriate. Thus, this manuscript can be recommended for publication in the Atmospheric Chemistry and Physics. There are, however, several places in the manuscript might be corrected to provide better readability.

Specific comments

Page 7 Line 194 I am wondering why the dry Hg deposition is increased in land areas? The authors should discuss more about it.

Page 15 Lines 400-403 The authors should disclose a data source or literatures to explain this part.

Page 15 Lines 410-412 The anomaly positive peaks were appeared in 2012 and 2014. Why does the volcanic emissions in 2011 might be responsible for higher GEM concentration in 2012? In addition, the authors should explain which volcanos were erupted in 2012 and 2014. Do these eruption or geothermal activities have impact on global GEM concentrations? I think more detailed discussion is needed in this part.

Technical corrections

Page 4 Figure 2 or Figure 3 The latitudinal and longitudinal values should be depicted in the Figure 2 or Figure 3 because the authors made the regionalization (Section 2.3) based on their values.

Page 5 Line 133 "However, he choise", I think this part has a typing mistake.

Page 5 Table 1 What is the unit of these numbers?

Page 7 Line 206 What type of known major Hg emitters does exist? In the conclusions section, the authors described a significant point source in South Africa is mostly linked to coal combustion. But they should give the information on the type of point source in the Section 2 or Section 3.

Page 12 Section 3.3 The authors explained the regionalized trends for GEM and other pollutant such as 222Rn and so on. I think a table or figure is needed in this part for better interpretation and understanding.
* * *
[Figure]

2020.

---

## Referee Comment (RC2) · Anonymous Referee #2 · 25 Mar 2020

The authors arrive at a number of, fortunately succinct, conclusions concerning the cycling of mercury in the southern hemisphere. While the conclusions themselves are not surprising, they are important because of the dearth of data from below the equator and therefore it is important that they are part of the scientific literature. I wholeheartedly recommend publication in ACP. That said, the manuscript is a little long-winded at times, some serious editing and perhaps moving some of the longer descriptions of the results to a Supplementary Information section would make the article more readable and accessible to the readers not familiar with the global mercury cycle. The conclusions could quite reasonably provide an outline for the structure of the manuscript which would make the article easier to read. The role of the South

[Figure]

African continental landmass and ocean regions and their associated characteristics (S. America: biomass burning/ASGM emissions, warm ocean (Agulhas- emissions), cold ocean (Antarctica- little variation)), for example. As it is, the manuscript provides a lot of information but little direction to where it is wants to arrive.

A few more detailed comments follow.

Abstract:

L17 'fulfill' maybe complete is better L21 the meaning of 'legacy emissions' may not be clear to all readers L21 when the authors use 'levels' are they referring to burdens, concentration fields, deposition fluxes, totals in environmental media? Last paragraph. The measured concentrations and trends from long-range transport are independent of the source region, but year to year variability is due to mining and biomass burning. These do not occur in Antarctica, the authors need to explain more clearly what they wish to say.

Introduction:

L43 'forces'? Maybe 'commits' or something similar would be more diplomatic. L58 could you add the Part 1 paper to the reference list

Methodology:

L86-7 The station has been .... 1970s L90 Paper 1 in the references Figure 1. As far as I am aware the emission inventory from the GMA 2018 is not in the public domain, so the authors need a private communication reference or to use the previous emission inventory, which is not dissimilar.

Modeling:

Figure 2. This needs to be revised. Readers areprobably not used to this projection and so it takes a while to realise we're looking at South America, South Africa and Antarctica.The scale is non-linear which is not mentioned. The dotted blue lines are

not clear, nor is the fact that they represent latitude and longitude.

Regionalization:

A figure showing where the regions are would be worth a thousand words. L133 the choice L134 how much does the uncertainty increacse? L142 Radon isn't anthropogenic, terrestrial maybe?

Identification of source/sink regions

Just a question, does the climate in South Africa divide into four seasons? L161 source/sink rather than just source regions L164 It might be more helpful to put this description in the caption figure, so that the figure is self-explanatory and reduce the descriptive text in the section. L173 low Radon shows less terrestrial influence, rather than less anthopogenic influence? Would it be possible to put quite a lot of this section into a series of bullet points/algorithm describing the procedure? Figure 3 needs a more descriptive caption.

Results:

If the authors stated their main findings and then described how the data/analysis supports these findings it would so much easier to read this section. Section 3.2 is the description of a table, very difficult to read. Section 3.3 L330 to L336 along with L349/50 sum up almost all that is necessary to say here, it seems. Section 3.4 If the authors removed L352 - L356.5 and started the section with 'Air masses from long-range transport . . . make up 90%' would it change anything apart from the readability? The Results should present Results. Preferably with short sentences to improve clarity. Section 3.5 This is extremely interesting, unfortunately the 'having shown . . ..' L388 is not convincing because the preceding sections are so prolix that the reader gets lost. L396 The identified processes are not stated clearly in the results section, or just not evident in all the extraneous description.

Conclusions:

Reading the article, the conclusions really were not made evident during the Results/Discussion (2) The Algulhas Current is mentioned twice in the article, and one of them is in the conclusions, so the reader should remember that the warm western boundary current flowing southwards between Madagascar and Mozambique is called Algulhas and influences atmospheric Hg concentrations in South Africa? I'm afraid the part that described its influence did not stand out in the Results. (3) So the measured Hg is of marine origin? (5) This is very interesting. How do the authors link gold mining though? The emission inventories we have do not cover these years, while biomass burning inventories are more up to date. Would it be possible to use a surrogate such as gold price which may indicate the profitability of artisanal and small scale gold mining. Industrial gold production is not an indicator of mercury emissions as the gold is refined using cyanide not mercury. (6) Very valid point, the ocean will mitigate mercury emission mitigation strategies for some time. The authors could highlight this, and also our current lack of knowledge of the details of the dynamics of ocean-atmosphere mercury exchange processes.

---

## Author Comment (AC1) · 24 Jun 2020

**Answers to reviewer #1**

**Page 7 Line 194 I am wondering why the dry Hg deposition is increased in land areas? The authors should discuss more about it.**

*The original wording is awkward and we rewrote this section. More more precise would be: The reason for this is probably a mixture of low emissions in the region and higher dry deposition on the terrestrial surface.*

**Page 15 Lines 400-403 The authors should disclose a data source or literatures toexplain this part.**

*We added a references:*

*Weigelt, A., Ebinghaus, R., Manning, A.J., Derwent, R.G., Simmonds, P., Spain, T.G., Jennings, S.G., Slemr, F.: Analysis and interpretation of 18 years of mercury observations since 1996 at Mace Head, Ireland, Atmospheric Environment 100, 85-93, 2015.*

*GMOS, Global Mercury Observation System, 2020. Available online: http://sdi.iia.cnr.it/geoint/publicpage/GMOS/gmos_monitor.zul*

**Page 15 Lines 410-412 The anomaly positive peaks were appeared in 2012 and 2014.Why does the volcanic emissions in 2011 might be responsible for higher GEM concen-tration in 2012? In addition, the authors should explain which volcanos were erupted in 2012 and 2014. Do these eruption or geothermal activities have impact on global GEM concentrations? I think more detailed discussion is needed in this part.**

After looking into detailed volcanic emission datasets we decided that the conclusion of a volcanic impact on mercury concentrations at Cape Point is too speculative. Therefore we removed this from our conclusions.

**Page 4 Figure 2 or Figure 3 The latitudinal and longitudinal values should be depictedin the Figure 2 or Figure 3 because the authors made the regionalization (Section 2.3)based on their values.**

We adjusted Figure 2 accordingly.
We added Figure 3 which depicts the source regions used for the analysis.

**Page 5 Line 133 "However, he choise", I think this part has a typing mistake.**

*We corrected the typo.*

**Page 5 Table 1 What is the unit of these numbers?**

*It is unitless. We added a description into the table caption:*
*"Total and relative allocation of trajectories to each source region depending on air parcel travel time."*

**Page 7 Line 206 What type of known major Hg emitters does exist? In the conclusions section, the authors described a significant point source in South Africa is mostly linked to coal combustion. But they should give the information on the type of point source in the Section 2 or Section 3.**

*We added the information that the major point sources are coal fired power plants.*

**Page 12 Section 3.3 The authors explained the regionalized trends for GEM and other pollutant such as 222Rn and so on. I think a table or figure is needed in this part forbetter interpretation and understanding.**

*We added additional tables to the manuscript.*

**Answers to reviewer #2**

*We want to thank the author, who surely is a native speaker of the english language, for his support in improving language and clarity of the paper.*

**Abstract:**

**L17 'fulfill' maybe complete is better**

*Done.*

**L21 the meaning of 'legacy emissions' may not be clear to all readers**

*We revised the abstract.*

**L21 when the authors use 'levels' are they referring to burdens, concentration fields, deposition fluxes, totals in environmental media?**

*We changed the abstract accordingly.*

**Last paragraph. The measured concentrations and trends from long-range transport are independent of the source region, but year to year variability is due to mining and biomass burning.These do not occur in Antarctica, the authors need to explain more clearly what they wish to say.**

*We clarified that we mean the variability observed at Cape Point:*

*Based on this dataset we were able to show that the inter-annual variability of Hg concentrations at Cape Point is not driven by climatology but rather due to changes in global emissions (gold mining and biomass burning).*

**Introduction:**

**L43 'forces'? Maybe 'commits' or something similar would be more diplomatic.**

*indeed*

**L58 could you add the Part 1 paper to the reference list**

*We added the reference*

**Methodology:**

**L86-7 The station has been.... 1970s**

*corrected*

**L90 Paper 1 in the references**

*corrected*

**Figure 1. As faras I am aware the emission inventory from the GMA 2018 is not in the public domain, so the authors need a private communication reference or to use the previous emission inventory, which is not dissimilar.**

*Indeed, the emission inventory is still not publicly available. We added a private communication reference in the caption.*

**Modeling:**

**Figure 2. This needs to be revised. Readers areprobably not used to this projection and so it takes a while to realise we're looking at South America, South Africa and Antarctica.The scale is non-linear which is not mentioned. The dotted blue lines are not clear, nor is the fact that they represent latitude and longitude.**

We adjusted Figure 2 accordingly.

[Figure]

**Regionalization:**

**A figure showing where the regions are would be worth a thousand words.**

*We added an additional Figure (#3) depicting the regions.*

**L133 the choice**
*corrected*

**L134 how much does the uncertainty increacse?**

*Altough it is not a linear process, as a rule of thumb I would consider the horizontal uncertainty of backward trajectories to be in the range of 100-200 km per day. A good read on the topic is: Engström and Magnusson, 2009*
*https://www.atmos-chem-phys.net/9/8857/2009/acp-9-8857-2009.pdf*

*We added this reference to the manuscript.*

**L142 Radon isn't anthro-pogenic, terrestrial maybe?**
*Agreed, we corrected the sentence.*

**Identification of source/sink regions Just a question, does the climate in South Africa divide into four seasons?**

*Yes it does. At -34°S it is roughly the same distance from the equator as Sicily and Crete in the Mediterranean.*

**L161source/sink rather than just source regions**

*corrected*

**L164 It might be more helpful to put this description in the caption figure, so that the figure is self-explanatory and reduce the descriptive text in the section.**

*We are reluctant to put such a long text into the figure caption. Especially as it is tought to give an introduction/explaination to Figures 3-7. If we would put it into the caption of Figure 4 it would probably be necessary to repeat it for the other figures.*

**L173 low Radon shows less terrestrial influence, rather than less anthopogenic influence? Would it be possible to put quite a lot of this section into a series of bullet points/algorithm describing the procedure?**

*Corrected*

**Figure 3 needs a more descriptive caption.**

*Figure 3: (This is now Figure 4) Distribution map for the 90th percentile highest $^{222}$Rn concentrations (left) and the 10th percentile lowest $^{222}$Rn concentrations (right) measured at Cape Point. Values are the dimensionless prevalence of air parcels of a given concentration percentile ranging from 0 to 1.This means that a homogeneous distribution of source ($90^{th}$ percentile) and sink ($10^{th}$ percentile) regions would lead to a plot with values of 0.1 everywhere. Deviations from this value indicate source and sink regions. See also the description in Section 2.4.*

**Results:**

**If the authors stated their main findings and then described how the data/analysis supports these findings it would so much easier to read this section.**

*We added a short paragraph at the begining of the results section giving an overview of our analysis and the major results. Moreover, we shortened Sections 3.2 and 3.3 to make it more readable/comprehendable:*

*„In this section we use backward trajectories of the $5^{th}$ and $95^{th}$ percentile GEM concentrations observed at Cape Point to identify the major source and sink regions for mercury (Section 3.1 "Source and sink regions"). We find that the eastern ocean with the warm Agulhas current is the major source region and the continent is the major sink region. We then compare the regional patterns of GEM with other pollutants (Section 3.2 "Comparison of regionalized data") and find that GEM shows a distinct pattern compared to pollutants of terrestrial, anthropogenic and photochemical origin. In Section 3.3 "Regional trends" we investigate distinct mercury trends for each region. We find that air masses from long range transport (South America, Antarctica) show no distinct trends which indicates that they are representative of the SH background. In Section 3.4 "Regional abundance" we investigate what impact changing atmospheric circulation may have on the GEM trend observed at Cape Point and find that it is negligible. Instead we find, that the annual average GEM concentrations depend on the regions with highest (estern ocean) and lowest (continental) GEM concentrations in air masses. Finally, in Section 3.5 "Inter-annual variability" we try to explain the inter-annual variability of GEM concentrations observed at Cape Point with changes in global emissions. We show that biomass burning and gold mining emissions can explain years with exceptionally high (2014) or low (2009) GEM concentrations."*

**Section 3.2 is the description of a table, very difficult to read.**

*We rewrote Sections 3.2 and 3.3 and added additional tables.*

**Section 3.3 L330 to L336 along with L349/50 sum up almost all that is necessary to say here, it seems.**

*We shortened this section albeit not as much as the reviewer asked for.*

**Section 3.4 If the authors removed L352 - L356.5 and started the section with 'Air masses from long-range trans-port...make up 90%' would it change anything apart from the readability? The Results should present Results. Preferably with short sentences to improve clarity.**

*We deleted the section as asked by the reviewer.*

**Section 3.5 This is extremely interesting, unfortunately the 'having shown....' L388 is not convincing because the preceding sections are so prolix that the reader gets lost.**

*We hope that the addition of a short introduction to the results section which states the major questions and findings furthers the readability of our paper. Moreover, we slightly shortened Sections 3.2 through 3.4 and added tables    for a better overview. There is still a lot of information in the results sections but we think that an interested reader might want to get some details on our study and readers not so interested in a certain analysis now have the chance to skip it and rely on the initial summary and the conclusions instead.*

**Conclusions:**

**Reading the article, the conclusions really were not made evident during the Results/Discussion**

*We rewrote parts of the Results Chapter (now called Results and Discussion) to address this issue.*

**(2) The Agulhas Current is mentioned twice in the article, and one of them is in the conclusions, so the reader should remember that the warm westernboundary current flowing southwards between Madagascar and Mozambique is called Algulhas and influences atmospheric Hg concentrations in South Africa? I'm afraid the part that described its influence did not stand out in the Results.**

*We now mention this more prominently at the beginning of Chapter 3 and in Section 3.1.*

**(3) So the measured Hg is of marine origin?**

*The Hg from the sector eastern ocean is mostly from marine sources (i.e. air-sea exchange). This underlines our finding (2) „The warm Agulhas current to the south-east is the major source of atmospheric mercury observed at Cape Point."*

**(5) This is very interesting. How do the authors link gold mining though? The emission inventories we have do not cover these years, while biomass burning inventories are more up to date. Would it be possible to use a surrogate such as gold price which may indicate the profitability of artisanal and small scale gold mining. Industrial gold production is not an indicator of mercury emissions as the gold is refined using cyanide not mercury.**

*For this we used ASGM Hg emission estimates based on world gold production as published by Streets et al., 2019. For the years 2016 and 2017 we used the data for 2015 as the dataset ends in 2015.*

[Figure]

Fig. 1. Estimates of annual Hg emissions (Mg) from Artisanal and Small-Scale Gold Mining (ASGM) obtained by trending the 2010 value from the GMA (UNEP, 2013; AMAP/UNEP, 2013) to other years in the period 2000–2015 using six alternative proxies. The average of all proxies is shown as the dotted red line. Also shown are the uncertainty ranges for the 2010 value estimated in this work and the GMA. (For interpretation of the references to colour in this figure legend, the reader is referred to the Web version of this article.)

*Fig. 1 from Streets et al., 2019*

**(6) Very valid point, the ocean will mitigate mercury emission mitigation strategies for some time. The authors could highlight this, and also our current lack of knowledge of the details of the dynamics of ocean-atmosphere mercury exchange processes.**

*We added a sentence highlighting this.*